# Optimizing Battery Charging Using Neural Networks in the Presence of Unknown States and Parameters

**DOI:** 10.3390/s23094404

**Published:** 2023-04-30

**Authors:** Andrea Pozzi, Enrico Barbierato, Daniele Toti

**Affiliations:** Department of Mathematics and Physics, Catholic University of the Sacred Heart, 25133 Brescia, Italy; enrico.barbierato@unicatt.it (E.B.); daniele.toti@unicatt.it (D.T.)

**Keywords:** machine learning, deep learning, neural networks, computational complexity, predictive control, battery management systems

## Abstract

This work investigates the effectiveness of deep neural networks within the realm of battery charging. This is done by introducing an innovative control methodology that not only ensures safety and optimizes the charging current, but also substantially reduces the computational complexity with respect to traditional model-based approaches. In addition to their high computational costs, model-based approaches are also hindered by their need to accurately know the model parameters and the internal states of the battery, which are typically unmeasurable in a realistic scenario. In this regard, the deep learning-based methodology described in this work was been applied for the first time to the best of the authors’ knowledge, to scenarios where the battery’s internal states cannot be measured and an estimate of the battery’s parameters is unavailable. The reported results from the statistical validation of such a methodology underline the efficacy of this approach in approximating the optimal charging policy.

## 1. Introduction

Managing Lithium-Ion batteries is a multifaceted undertaking that necessitates the careful negotiation of various factors and approaches, which may frequently be at odds with one another. This involves defining appropriate current profiles for both charging and discharging the batteries, implementing mechanisms to mitigate performance degradation over time, and ensuring their safety throughout their lifespan. Given the breadth of these considerations, it is evident that managing Lithium-Ion batteries is a challenging task that requires meticulous attention [1]. Battery management systems used in the industry usually rely on rule-based algorithms. One widely used class of algorithms is based on the constant-current/constant-voltage principle, which seeks to optimize the charging time of the batteries while staying within the voltage limits that have been established [2]. Despite their widespread use, these mechanisms tend to employ a conservative approach that often fails to fully leverage the potential of Lithium-Ion batteries, whether it be to expedite the charging time or safeguard their safety. Consequently, the outcomes of these algorithms are often suboptimal. Moreover, due to the fixed nature of voltage constraints, such basic algorithms overlook the gradual deterioration of batteries and the progressive alteration of their internal characteristics as they undergo numerous cycles of charging and discharging [3].

In reality, there are methods available that can offer a more effective control strategy by exploiting a mathematical model of the battery. Within this context, different control approaches have been considered, such as fuzzy logic [4,5], empirical rules [6,7], and optimization-based strategies [8,9]. Among the latter, one such noteworthy technique is model predictive control (MPC) [10], which has achieved considerable success and widespread adoption in the context of battery management (see for instance [11,12,13,14,15,16,17,18]). This is owing to its ability to manage complex, nonlinear processes involving multiple variables while satisfying constraints related to inputs and states. Despite its effectiveness in simulations, MPC is not without limitations. Its practical use in real-world scenarios is often impeded by computational complexities, which can render it less efficient and effective than its theoretical potential suggests, especially when accurate nonlinear models of the system are employed. This is due to the nature of the algorithm, which must solve a constrained optimal control problem at each time step in real time. When it is not able to computationally be on par with a fast sampling time of the control law, MPC becomes useless.

To address this challenge, researchers have proposed a potential solution in the form of an explicit MPC algorithm, which has been applied in the context of battery charging by the authors in [19]. This type of MPC, as introduced in [20], aims to reduce the computational burden of real-time operations by simplifying them to the evaluation of a straightforward function. In theory, the explicit MPC should require fewer computations since it precomputes the optimal control action in the form of a piece-wise function of the state and reference vectors, and only needs to detect the region in which the states are located during real-time operations. However, in practice, the detection of such a region can be computationally demanding, especially when dealing with numerous constraints and long prediction horizons, leading to performance degradation that may not be acceptable [21].

The computational cost of the explicit MPC has been a challenge, leading to several works in the literature attempting to address the issue (for instance, [22,23,24]). Most of these works propose an approximation of the control law, with machine learning models gaining attention in recent years, giving birth to the concept of learning-based model-predictive control. Among the various learning models, deep neural networks have been successful, leading to the development of the so-called deep MPC. Previous works in deep MPC include [25], where the authors describe a model that is robust to input errors, and [21], where a deep predictive controller is shown to be able to exactly represent—with a sufficiently high number of neurons and layers—an explicit MPC control law. As far as the battery control is concerned, a deep MPC formulation has been proven successful in mitigating the issues of the computational complexity of a standard predictive controller in [26,27], where a learning-based charging approach is applied to a battery modeled as an equivalent circuit model and as an electrochemical one, respectively.

Beyond the problem of computational complexity, which can be reduced with the use of approximations based on neural networks, the use of a predictive control paradigm within the context of battery management is also limited by the fact that it relies on the following assumptions: (i) the availability of an accurate model of the battery dynamics (with accurately estimated parameters), (ii) measurability of all the relevant states of the system. Such assumptions hold only in an ideal scenario in which the controller has a perfect knowledge of the system, while in practice the parameters and the internal states of the system need to be inferred from the available measurements (voltage, current, and surface temperature) and only the model structure can be assumed to be known a priori. The problem of parameter and state estimations for Lithium-Ion batteries has been largely discussed in the literature and it is usually considered a key issue in the development of reliable controllers. The proposed solutions usually involve the execution of ad hoc experiments to collect the data required for the estimation of the parameters (see [28]) and the design of suitable observers to reconstruct the state trajectory online (see [29]). However, it is worth highlighting that the accuracy of the state observer is highly related to that of the estimated electrochemical parameters, which can vary greatly, even among cells of the same type, and may change as the battery ages. In conclusion, determining these parameters with the required level of accuracy often needs time-intensive and intrusive experiments.

It is clear that a deep MPC approach that simply approximates an MPC feedback law, such as the one proposed in [26,27], still suffers from the aforementioned issues related to state/parameter estimation. In this paper, an extension is proposed for the work presented in [27], with the aim of considering a more realistic scenario in which only the current, voltage, and temperature are assumed to be measurable, and the battery parameters are unknown. Specifically, the implementation of an output-based deep MPC is considered here, which is able to charge the battery with the knowledge of the system restricted to the model structure, which is considered here as that of the well-known single particle model (SPM) [30]. The results show that the proposed approach can accurately reproduce the benchmark performance in a practical scenario.

For the first time, to the knowledge of the authors, a computationally-efficient neural network-based predictive controller was successfully employed for the optimal charging of a Lithium-Ion battery in the presence of non-measurable states and unknown parameters.

The structure of this work is as follows: in Section 2, the battery model is presented in detail. Section 3 presents the formulation of the learning-based predictive controller. Section 4 shows the training approach. The experimental results are shown in Section 5, while Section 6 discusses potential improvements to the proposed approach. Finally, Section 7 provides the concluding remarks.

## 2. Model

The primary categories of models used in sophisticated battery management systems (BMSs) include equivalent circuit models (ECMs) [31,32] and electrochemical models (EMs) [33,34]. ECMs are relatively simple and intuitive, whereas EMs offer a comprehensive explanation of the electrochemical processes occurring within a cell. Electrochemical models are more appropriate for simulation purposes rather than for real-time control applications. Furthermore, the implementation of electrochemical models in a control framework is constrained by issues of identifiability and observability [35]. As a result, researchers have been focusing on the creation of simplified electrochemical models that are quicker to simulate, identifiable, observable, and still provide adequate representation of the internal cell phenomena [36,37]. The single-particle model (SPM) [30], which is obtained from the pseudo-two-dimensional model [38] by considering the two electrodes as spherical particles, is one notable example among these models. SPM is used in this paper to mathematically describe the battery dynamics. Such a simplified electrochemical model has been largely adopted for battery control and estimation of the states, due to its ability in achieving a reasonable trade-off between the computational cost and accuracy (see e.g., [27,39,40,41]). The accuracy of such a model was demonstrated in [42], among others. Note that the battery model is enriched with the two-state temperature dynamics proposed by the authors in [43] to account for thermal phenomena.

Only the equations pertaining to the primary variables of the model are mentioned below; for a more comprehensive explanation, the reader is directed to reference [27]. Specifically, the variable soc(t)∈[0,1] represents the state of charge of the battery, whose temporal evolution is given by:(1)$$\begin{array}{c} \frac{d\;\mathrm{soc}(t)}{d\;t}=\frac{I(t)}{3600C} \end{array}$$
where the applied current is denoted as I(t), with the convention that a positive current charges the cell and *C* represents the cell capacity in [Ah]. It is important to note that when the battery is fully charged, the state of charge is at soc(t)=1, and when completely discharged, the state of charge is at soc(t)=0. Moreover, the battery voltage is given by the following equation:(2)$$\begin{array}{c} V\left(t\right)=U_{p}\left(t\right)-U_{n}\left(t\right)+\eta_{p}\left(t\right)-\eta_{n}\left(t\right)+R_{sei}I\left(t\right) \end{array}$$
where the terms $U_{i}\left(t\right)$ and $\eta_{i}\left(t\right)$, for i∈{n,p}, represent the open circuit potential and overpotential, respectively, as defined in Section 2 of [27], while the term $R_{sei}I\left(t\right)$ describes the voltage drop in the solid electrolyte interphase (SEI) resistance. Note that the open circuit potentials and the overpotential are nonlinear functions of the applied current, state of charge, and battery average temperature. As far as the latter is concerned, the two-state model proposed in [43] is adopted here for the thermal dynamics, in which the core and the surface temperatures are represented by $T_{c}\left(t\right)$ and $T_{s}\left(t\right)$, respectively. In particular, it holds that:
(3a)$$\begin{array}{cc} C_{c}\frac{d\;T_{c}\left(t\right)}{d\;t}= & \;Q\left(t\right)-\frac{T_{c}\left(t\right)-T_{s}\left(t\right)}{R_{c,s}} \end{array}$$
(3b)$$\begin{array}{cc} C_{s}\frac{d\;T_{s}\left(t\right)}{d\;t}= & \;\frac{T_{c}\left(t\right)-T_{s}\left(t\right)}{R_{c,s}}-\frac{T_{s}\left(t\right)-T_{env}}{R_{s,e}} \end{array}$$
where $R_{c,s}$ and $R_{s,e}$ denote the thermal resistances between the core and surface and between the surface and the external environment, respectively, whereas $C_{c}$ and $C_{s}$ represent, respectively, the heat capacity of the cell’s core and surface. Finally, Q(t) represents the amount of heat generated, which is defined as follows:(4)$$\begin{array}{cc} Q(t)= & |I\left(t\right)(V\left(t\right)-U_{p}\left(t\right)+U_{n}\left(t\right))|. \end{array}$$

It is important to highlight that the electrochemical parameters in nominal form have been extracted from the experimental characterization of a commercial cell, specifically the Kokam SLPB 75106100, as presented in [44,45], while the thermal ones are based on those employed by [43].

## 3. Methodology

This section outlines the methodology behind the learning-based predictive controller. First of all, the nonlinear equations that describe the plant dynamics are formulated in a state space form and discretized as follows:
(5a)$$\begin{array}{cc} x(t_{k+1})= & f_{d}\left(x\left(t_{k}\right),\;u\left(t_{k}\right),\;p\right) \end{array}$$
(5b)$$\begin{array}{cc} y(t_{k+1})= & g(x\left(t_{k+1}\right),\;u\left(t_{k}\right),\;p) \end{array}$$
in order to consider a digital controller that applies a piece-wise constant input at the discrete times $t_{k},\;k\in N$ with sample time $t_{s}$. Specifically, $x\left(t_{k}\right)\in R^{n_{x}}$, $u\left(t_{k}\right)\in R^{n_{u}}$, and $y\left(t_{k}\right)\in R^{n_{y}}$ represent the state, input, and output vectors, respectively. Moreover, the model parameters are represented as $p\in R^{n_{p}}$ (the dependence of a variable on the parameter vector will be made explicit below only where necessary). Furthermore, the functions $f_{d}:R^{n_{x}}\times R^{n_{u}}\to R^{n_{x}}$ and $g:R^{n_{x}}\times R^{n_{u}}\to R^{n_{y}}$ map the current state and the current input into the next state ($x(t_{k+1})$) and the output ($y(t_{k})$), respectively. Finally, note that the generic input sequence applied in the time interval $\left[t_{k},\;t_{k+H}\right]$, with H∈N, is represented as:(6)$$\begin{array}{c} u_{\left[t_{k},\;t_{k+H}\right]}={\left[u{\left(t_{k}\right)}^{⊤}\;u{\left(t_{k+1}\right)}^{⊤}\;\cdots \;u{\left(t_{k+H-1}\right)}^{⊤}\right]}^{⊤}. \end{array}$$

The rest of the section is organized as follows. The main features of a standard predictive controller are discussed in Section 3.1, while its approximation through a learning-based algorithm under the assumption of state measurability and parameter knowledge is presented in Section 3.2. Finally, in Section 3.3, * a novel algorithm is proposed, which still relies on neural networks and consists of the adaptation of the deep MPC for the case in which the states and parameters are unknown.

### 3.1. Model Predictive Control

Predictive control methodologies that rely on a receding horizon framework [46] have been shown to be particularly effective in dealing with nonlinear processes subject to the input and state constraints [10]. In this context, the model’s predictive control scheme computes the optimal control sequence $u_{\left[t_{k},\;t_{k+H}\right]}^{★}$ over the prediction horizon *H* at each time step $t_{k}$, by solving a constrained optimization problem with a cost function that depends on predictions made by a mathematical model of the plant. Then, according to the receding horizon paradigm, only the first element $u^{★}\left(t_{k}\right)$ of the resulting optimal input sequence is applied, while the remaining future optimal moves are discarded.

Specifically, the following optimization problem is solved at each time step in order to compute the control action:(7)$$\begin{array}{c} u_{\left[t_{k},t_{k+H}\right]}^{★}=\underset{u_{\left[t_{k},t_{k+H}\right]}}{\mathrm{argmin}}J\left(x\left(t_{k}\right)\right) \end{array}$$
subject to:
(8a)$$\begin{array}{c} \mathrm{system}\;\mathrm{dynamic}\;\mathrm{in}\;(5) \end{array}$$
(8b)$$\begin{array}{cc} u^{lb}\leq u\left(t_{i}\right)\leq u^{ub},\; & i=k,\;k+1,\;\cdots ,\;k+H-1 \end{array}$$
(8c)$$\begin{array}{cc} x^{lb}\leq x\left(t_{i}\right)\leq x^{ub},\; & i=k+1,\;k+1,\;\cdots ,\;k+H \end{array}$$
(8d)$$\begin{array}{cc} y^{lb}\leq y\left(t_{i}\right)\leq y^{ub},\; & i=k+1,\;k+1,\;\cdots ,\;k+H \end{array}$$
with $u^{lb},\;u^{ub}\in R^{n_{u}}$ being the limits for the input vector, $x^{lb},\;x^{ub}\in R^{n_{x}}$ the ones for the state vector, and $y^{lb},\;y^{ub}\in R^{n_{x}}$ the ones for the output vector. The cost function $J(x\left(t_{k}\right))$ to be minimized is formulated as follows:(9)$$\begin{array}{c} \begin{array}{cc} J(x\left(t_{k}\right))= & \;∥x\left(t_{k+H}\right)-x_{\mathrm{ref}}{∥}_{Q_{H}}^{2}+\sum_{i=k+1}^{k+H}\left[∥x\left(t_{i}\right)-x_{\mathrm{ref}}{∥}_{Q_{x}}^{2}+{∥y\left(t_{i}\right)-y_{\mathrm{ref}}∥}_{Q_{y}}^{2}\right]+\\ & +\sum_{i=k}^{k+H-1}{∥u\left(t_{i}\right)-u_{\mathrm{ref}}∥}_{R}^{2} \end{array} \end{array}$$
where the vectors $x_{\mathrm{ref}}\in R^{n_{x}}$, $y_{\mathrm{ref}}\in R^{n_{y}}$ and $u_{\mathrm{ref}}\in R^{n_{u}}$ correspond to the reference point that the MPC aims to track and the matrices $Q_{x}\in R^{n_{x}\times n_{x}}$, $Q_{y}\in R^{n_{y}\times n_{y}}$,$Q_{H}\in R^{n_{x}\times n_{x}}$, and $R\in R^{n_{u}\times n_{u}}$ are design parameters, with $Q_{x},\;Q_{y},\;Q_{H}\geq 0$ and R>0. Note that the term $∥x\left(t_{k+H}\right)-x_{\mathrm{ref}}{∥}_{Q_{H}}^{2}$ represents a suitably tuned terminal penalty used to improve the controller stability.

The MPC control law is defined as follows:(10)$$\begin{array}{c} u_{\mathrm{mpc}}\left(x\left(t_{k}\right)\right)=u{\left(t_{k}\right)}^{★} \end{array}$$
which is computed by solving the problem in (Equation 7) for the state vector $x(t_{k})$.

### 3.2. State-Based Deep MPC

Learning-based predictive control leverages the representation capabilities of machine learning models to achieve a precise approximation of the feedback law used in standard predictive controllers. This approach offers an alternative to the concept of “explicit” MPC, with the goal of further reducing the online computational burden of the control scheme. Explicit MPC relies on a piece-wise function to represent the control law, which requires finding the polytopic region where the states are located online. This process can be computationally expensive, especially if the number of regions that define the corresponding piece-wise function is high. In contrast, learning-based methodologies solve this issue by directly mapping the current states to the optimal action using a machine learning model.

The notion of learning-based MPC was initially introduced by [47]. However, its practical application in designing fast MPC techniques has only become reliable recently, primarily due to new advancements in the theoretical description of neural networks. In the following part of this section, the deep MPC formulation used in this study is presented.

A deep learning model N=N(x,r,θ) with $n_{x}$ inputs and $n_{u}$ outputs can be described in this fashion, where θ stands for the parameters of the data-driven model, whereas r represents the vector of the MPC references, i.e., $r={\left[x_{\mathrm{ref}}^{⊤}\;y_{\mathrm{ref}}^{⊤}\;u_{\mathrm{ref}}^{⊤}\right]}^{⊤}$. Synthetic data for training are produced by solving Equation (Equation 7) for $n_{\mathrm{tr}}$ different reference samples and different states, i.e., $r_{i}$ and $x_{\mathrm{tr},i}$, respectively, for $i=1,\;2,\;\ldots ,\;n_{\mathrm{tr}}$, and by saving the tuples $\left(x_{\mathrm{tr},i},\;r_{i},\;u_{\mathrm{mpc}}\left(x_{\mathrm{tr},i}\right)\right)$ in the dataset $B_{\mathrm{tr}}$, where $u_{\mathrm{mpc}}\left(\cdot \right)$ represents the MPC control action.

The model denoted by N is subjected to offline training using the dataset $B_{\mathrm{tr}}$ via the backpropagation method to resolve the following optimization problem:(11)$$\begin{array}{c} \theta^{★}=\underset{\theta }{\mathrm{argmin}}\frac{1}{n_{\mathrm{tr}}}\sum_{i=0}^{n_{\mathrm{tr}}}{∥u_{\mathrm{mpc}}\left(x_{\mathrm{tr},i}\right)-N\left(x_{\mathrm{tr},i},\;r_{i},\;\theta \right)∥}_{2}^{2} \end{array}$$
where the loss function is the mean squared error between the model prediction and the target. Finally, the actual control action is obtained as follows:(12)$$\begin{array}{c} u_{s-\mathrm{dmpc}}\left(x\left(t_{k}\right)\right)=N\left(x\left(t_{k}\right),\;r,\;\theta^{★}\right) \end{array}$$
where the subscript “s-dmpc” indicates the state-based deep MPC algorithm.

### 3.3. Output-Based Deep MPC

As previously stated, the reliability in a practical scenario of the algorithm proposed in Section 3.2 is limited by the assumption of full-state measurability. Within this context, it is worth noticing that the design and tuning of a suitable state observer is usually considered a time-consuming process that still relies on the strong assumption of knowledge of the model parameters. In a realistic situation, the latter can only be estimated with a certain degree of accuracy, which may significantly affect the performance of the state observer. Moreover, inaccuracies in the parameters may lead to wrong predictions of the system’s state evolution, thus making pointless the predictive nature of the controller. As a possible solution, a novel methodology is proposed here, which relies only on the available measurements to approximate the optimal control action. In particular, inspired by the techniques used in the context of the partially observable Markov decision process, a fixed window of historical measurements is used as input for the neural network during the training phase, thus allowing the model to learn a map from the available measurements to the optimal action.

Therefore, the training phase is firstly reformulated by relying on a dataset $B_{\mathrm{tr}}^{out}$, with $|B_{\mathrm{tr}}^{out}|=n_{tr}-n_{w}$, in which the *i*-th sample consists of a feature vector $f_{i}$ and the target $u_{\mathrm{mpc}}\left(x_{\mathrm{tr},i}\right)$, with the former defined as follows:(13)$$\begin{array}{c} f_{i}=\left[y_{\mathrm{tr},i-n_{w}},\;\ldots ,\;y_{\mathrm{tr},i-1},\;u_{\mathrm{tr},i-n_{w}},\;\ldots ,\;u_{\mathrm{tr},i-1},\;r_{i}\right] \end{array}$$
where $y_{\mathrm{tr},i}$ and $u_{\mathrm{tr},i}$ are the samples of the system’s outputs and inputs, respectively, while $n_{w}$ is the length of the window of the considered historical measurements. The neural network N is then trained offline on the dataset $B_{\mathrm{tr}}^{out}$ by solving the following optimization through the backpropagation method:(14)$$\begin{array}{c} \theta_{out}^{★}=\underset{\theta }{\mathrm{argmin}}\frac{1}{n_{\mathrm{tr}}-n_{w}}\sum_{i=n_{w}}^{n_{\mathrm{tr}}}{∥u_{\mathrm{mpc}}\left(x_{\mathrm{tr},i}\right)-N\left(f_{i},\;\theta \right)∥}_{2}^{2} \end{array}$$
where the loss function is the mean squared error between the network prediction and the target. The control action is obtained from the following equation:(15)$$\begin{array}{c} u_{o-\mathrm{dmpc}}\left(x\left(t_{k}\right)\right)=N\left(f_{i},\;\theta_{out}^{★}\right) \end{array}$$
where the subscript “o-dmpc” indicates the output-based deep MPC algorithm.

The proposed output-based deep MPC relies on the idea that a sufficiently large window of historical measurements contains the information necessary to reconstruct the state of the system, and that the neural network is able to approximate the optimal action to take in a particular state by recognizing its pattern in the available measurements. However, the possibility of using the presented technique in a realistic scenario strictly depends on how the training set $B_{\mathrm{tr}}^{out}$ is generated. For instance, if the model’s parameters are kept constant through the whole generation procedure, the algorithm is expected to achieve high performance only if the parameters of the system coincide with the ones used to generate the training set. However, as mentioned above, the assumption of accurate knowledge of the model’s parameters has limited practical support. For this reason, an adaptation of the output-based deep MPC is presented here with the aim of making the control algorithm robust to variations in the parameters of the system. Specifically, such a result is achieved by generating each sample *i* of the training dataset $B_{\mathrm{tr}}^{out}$ by integrating the system dynamics in (5) and solving the optimization problem (Equation 7) with a different value of the parameter vector $p_{i}$. As a consequence, the training of the neural network becomes
(16)$$\begin{array}{c} \theta_{out}^{★}=\underset{\theta }{\mathrm{argmin}}\frac{1}{n_{\mathrm{tr}}-n_{w}}\sum_{i=n_{w}}^{n_{\mathrm{tr}}}{∥u_{\mathrm{mpc}}\left(x_{\mathrm{tr},i},\;p_{i}\right)-N\left(f_{i},\;\theta \right)∥}_{2}^{2} \end{array}$$
which is based on the idea that a deep learning model can learn the optimal action to take in a particular state and parameter configuration by searching for a specific pattern in the available historical measurements.

## 4. Training of the Controller

This section is devoted to the definition of the optimal charging problem for a Lithium-Ion cell in Section 4.1, as well as the dataset generation and model training phases in Section 4.2 and Section 4.3, respectively.

### 4.1. Optimal Battery Charging

Within this section, a problem of optimal control is presented concerning the fast charging of a battery. The main objectives are to track the state of charge while reducing the current flow, with the additional constraint of maintaining safe voltage and temperature levels. The battery dynamics are modeled as a nonlinear and discrete system where the current applied to the battery acts as the input variable. Finally, the optimization problem is formulated according to the following procedure:(17)$$\begin{array}{cc} \min_{I_{\left[t_{k},t_{k+H}\right]}} & q_{\mathrm{soc}}\sum_{i=k+1}^{k+H}{\left(\mathrm{soc}\left(t_{k}\right)-{\mathrm{soc}}_{\mathrm{ref}}\right)}^{2}+r\sum_{i=k}^{k+H-1}I{\left(t_{k}\right)}^{2}+\\ \; & +q_{H}{\left(\mathrm{soc}\left(t_{k+H}\right)-{\mathrm{soc}}_{\mathrm{ref}}\right)}^{2} \end{array}$$
that for i=k,k+1,⋯,k+H−1 are subject to:
(18a)$$\begin{array}{c} \mathrm{battery}\;\mathrm{dynamics}\;\mathrm{in}\;(1)–(4) \end{array}$$
(18b)$$\begin{array}{c} 0\leq I(t_{i})\leq 10\;A \end{array}$$
(18c)$$\begin{array}{c} 0\leq \mathrm{soc}(t_{i})\leq 1 \end{array}$$
(18d)$$\begin{array}{c} T_{c}\left(t_{i}\right)\leq 313.15\;K \end{array}$$
(18e)$$\begin{array}{c} T_{s}\left(t_{i}\right)\leq 313.15\;K \end{array}$$
(18f)$$\begin{array}{c} V(t_{i})\leq 4.2\;V \end{array}$$
with $q_{\mathrm{soc}}=1$, $r=10^{-6}$ and $q_{H}=1$. Moreover, the length of the prediction horizon is chosen as H=4 and the sample time is taken as $t_{s}=10\;s$. Interestingly enough, the reference state of charge (${\mathrm{soc}}_{\mathrm{ref}}$) is left unspecified due to its dependence on charging preferences. This is in contrast to the current, temperature, and voltage limits, which are determined solely by the battery chemistry. During the dataset generation phase for the learning-based algorithm, the reference state of charge is modeled as a random variable with a uniform probability distribution within a designated range. This enables the learning-based MPC to adapt the charging profile to the various potential values of the reference. Finally, the input reference, which is the current reference, is always set to $I_{\mathrm{ref}}=0$ since the system is marginally stable.

### 4.2. Dataset Generation

In order to train the learning-based algorithm effectively, it is necessary to have a large dataset that includes information on the system’s states and parameters, as well as the optimal action taken by the predictive controller. The quality of the data is critical, as it can significantly affect the performance of the proposed data-driven algorithm. Having inaccurate, inconsistent, or flawed data can have a negative impact on the performance of a data-driven algorithm, such as the one presented in this manuscript, potentially leading to incorrect or biased results, poor performance, and decreased trust in the algorithm and its outcomes. Furthermore, the quantity of data used is also crucial in avoiding overfitting in a machine learning model. Overfitting takes place whenever a model ends up being too complex and includes too many parameters with respect to the amount of training data; this phenomenon makes it so that the model memorizes the training data, with the consequence of displaying a poor performance when applied to new, previously unseen data. Overfitting can be alleviated by increasing the amount of training data, so that the model is provided with more examples to learn from and, thus, may be able to generalize to new data. Even so, a mere increase in the quantity of training data may not be enough to prevent the overfitting phenomenon since the characteristics of the training data used need to be considered. In fact, it is important for those data to be as varied and representative as possible with respect to their real-world distribution, in order for the model not to memorize specific patterns or features in the data. In this regard, robust models for machine learning, with a high potential for generalization when dealing with unseen data, need to be fed both high-quality and large quantities of data.

As a result, firstly, an ideal MPC controller that executes the control task specified in Section 4.1 is implemented. To produce the dataset, a synthetic method is utilized, involving the offline execution of 2000 simulations, each containing 200 time steps. For each sample time *i*, a dataset sample is created, including the available measurements, the reference for the state of charge (determined for the *j*-th simulation), and the optimal current value for the subsequent time step:(19)$$\begin{array}{c} \left(V_{i-1},\;T_{s,i-1},\;I_{i-1},\;{\mathrm{soc}}_{\mathrm{ref},j},\;I_{i}^{★}\right). \end{array}$$

In order to include a window of historical measurements of size $n_{w}$, the database is reshaped with the rows as follows:(20)$$\begin{array}{c} \left(V_{i-n_{w}},\;T_{s,i-n_{w}},I_{i-n_{w}},\;\ldots ,\;V_{i-1},\;T_{s,i-1},\;I_{i-1},\;{\mathrm{soc}}_{\mathrm{ref},j},\;I_{i}^{★}\right) \end{array}$$
i.e., the previous measurements are used to predict the optimal current. Note that for each simulation *j*, the battery’s initial conditions are extracted from a random uniform distribution: $\mathrm{soc}\left(t_{0}\right)∼U\left(0,\;1\right)$ and $T_{s}\left(t_{0}\right),\;T_{c}\left(t_{0}\right)∼U\left(298.15\;K,\;313.15\;K\right)$. Similarly, the reference state of charge for the *j*-th simulation is sampled as ${\mathrm{soc}}_{\mathrm{ref},j}∼U\left(0.7,\;1\right)$. Finally, in order to make the algorithm robust to changes in the parameters during the battery aging, the battery capacity and the SEI resistance are sampled from a uniform distribution, thus obtaining a different parameter vector $p_{j}$ in every simulation: $C_{j}∼U\left(5.5\;Ah,\;8\;Ah\right)$ and $R_{sei,j}∼U\left(0.014\;\Omega ,\;0.019\;\Omega \right)$. Note that the pair C=8Ah and $R_{sei}=0.014\;\Omega$ corresponds to that of a brand new battery just released from the factory, while C=5.5Ah and $R_{sei}=0.019\;\Omega$ represents a battery at the end of its life.

To reiterate, it is crucial to remember that for each step *i*, the optimal current $I_{i}^{★}$ is determined by solving the optimization problem described in (Equation 17). This process involves executing an ideal MPC, assuming full-state measurability and complete knowledge of the relevant parameters. To increase exploration, Gaussian noise is added to the MPC control action during the battery dynamics evolution for every simulation, using a standard deviation of 2,A. Lastly, the dataset is split into three separate sets, as is customary in machine learning: training, validation, and testing.

### 4.3. Training Phase and Model Selection

In this section, the training phase of several machine learning configurations is conducted, with a particular focus on feed-forward deep neural networks (DNNs) and recurrent neural networks (RNNs). As can be seen from Table 1, which reports the results of the training phase for the different architectures and measurement window $n_{w}$, the use of RNNs appears to be particularly suitable for solving the considered task, thanks to their ability in dealing with sequential information, such as the one contained in the dataset samples (see (Equation 20)). Note that all of the considered models have a *ReLu* activation function in the hidden layers and a *tanh* activation function in the output layer. The *tanh* activation function on the last layer is used to constrain the output of the network within a specific range, which means, in the context of battery charging, constraining the optimally applied current within its range of operation. In order to enhance the learning capabilities of all of the considered models, a preprocessing pipeline is considered that involves both the scaling and standardization of the dataset’s features. Moreover, for the training of the deep learning models, the stochastic gradient descent algorithm is employed, and, in particular, the *Adam* optimizer is adopted, with the mean squared error as the loss function and learning rate equal to $5\times 10^{-4}$. Finally, it is worth highlighting that Gaussian noise was applied to the features in the training set with the dual motivation of preventing any overfitting and making the model robust to a realistic scenario, in which the values of surface temperature and voltage were affected by a measurement disturbance due to malfunctioning or inaccurate sensors. Specifically, Gaussian noises with standard deviations of 20 mV for the voltage and 1 K for the temperature were considered.

The model that achieved the highest performance consisted of a recurrent neural network with 4 long short-term memory (LSTM) hidden layers (with 128, 64, 32, and 16 neurons each) and 4 fully connected hidden layers (2 of them with 100 neurons each, one with 50 neurons, and the last one with 10 neurons), with a window of historical measurements $n_{w}=20$. Therefore, such a model was selected to properly approximate the predictive control law in the simulations considered in Section 5.

It is important to note that the loss of validation for the selected model converged after 23 epochs, as depicted in Figure 1, and that no overfitting was present due to the fact that the prediction error during the testing phase ($e_{\mathrm{test}}=0.109$) was coherent with that achieved on the training set.

## 5. Results

This section illustrates the results of the comparison, regarding the case of battery charging, between an ideal predictive controller, which operates under the assumptions of state measurability and parameter knowledge, and an output-based deep MPC, for which only the model structure is assumed to be known a priori. More specifically, the effectiveness of the proposed algorithm in tracking the optimal charging profile for a battery with uncertain parameters by exploiting only noisy measurements is demonstrated in Section 5.1, for the first time, to the knowledge of the authors. Finally, in Section 5.2, details on the software implementation are provided.

### 5.1. Approximation of the Optimal Charging Profile in the Presence of Unknown States and Parameters

In the following subsection, the effectiveness of the neural network-based approach in approximating the charging profile achieved by the ideal predictive controller will be evaluated, with particular attention to different combinations of initial conditions, references, and battery parameters. The results are illustrated in Figure 2, Figure 3, Figure 4, Figure 5 and Figure 6, with crossed lines representing the output-based deep MPC profiles and solid lines representing the ideal predictive controller. Specifically, the figures on the left refer to an almost new battery with parameters C=7.5 Ah and $R_{sei}=0.015\;\Omega$, while the ones on the right represent an aged battery with C=6 Ah and $R_{sei}=0.018\;\Omega$. Moreover, two scenarios of initial conditions and state-of-charge references are considered: blue lines are used for the profiles, which refer to the simulation with ${\mathrm{soc}}_{\mathrm{ref}}=0.8$ and initial states given by $\mathrm{soc}(t_{0})=0.2$, $q_{n}\left(t_{0}\right)=0$, $q_{p}\left(t_{0}\right)=0$ and $T_{s}\left(t_{0}\right)=T_{c}\left(t_{0}\right)=305.15$ K (also referred as *sim 1*), while the red lines are used for the case of ${\mathrm{soc}}_{\mathrm{ref}}=1$ and initial states given by $\mathrm{soc}(t_{0})=0.05$, $q_{n}\left(t_{0}\right)=0$, $q_{p}\left(t_{0}\right)=0$, and $T_{s}\left(t_{0}\right)=T_{c}\left(t_{0}\right)=300.15$ K (*sim 2*). For all of the simulations, a prediction horizon H=4 is considered, equal to the one used for the generation of the training dataset.

It is important to recall that in every simulation the output-based deep MPC does not have access to the battery states and is not aware of the value of the battery parameters, but it relies only on the available measurements, which are assumed to be affected by a Gaussian noise with a standard deviation of 20 mV for the voltage and 1 K for the surface temperature.

As seen from the results, the trajectory of the system controlled by the proposed algorithm exhibits characteristics that are very similar to those achieved by using the ideal MPC. It is important to note that the ideal MPC is based on strong assumptions, such as full-state measurability and known parameters, which may not hold in a realistic scenario. Therefore, it is considered as a benchmark in this study. In particular, Figure 2 demonstrates that both controllers are capable of tracking the state of charge references in all considered situations. Moreover, Figure 3, Figure 4 and Figure 5 show that the output-based deep MPC is almost always able to satisfy the constraints on voltage and temperature. Furthermore, in Figure 6, the applied current is shown, where the constraints are satisfied by the design of the output-based deep MPC, while in the case of the ideal predictive controller, they are imposed as bounds on the input. The control strategy obtained with both the controllers for *sim 2* is similar to the constant-current/constant-temperature/constant-voltage protocol, which has been proven to be effective in decreasing the charging time while adhering to the current, voltage, and temperature restrictions (see [48,49]).

It should be noted that in realistic scenarios, such as the one considered for the proposed algorithm, slight violations of voltage and temperature constraints may occur due to unknown internal states, unknown parameters, and measurement noises affecting the available output. In practical situations, it is not always feasible to ensure strict satisfaction of hard constraints on states and outputs. Instead, these constraints should be considered soft constraints, meaning they should be avoided if possible. Appropriate measures can be taken to minimize the likelihood of a constraint violation. For instance, using a more accurate machine learning model in output-based deep MPC may help reduce constraint violations. As for the satisfaction of input constraints, the inclusion of a *tanh* activation function on the output layer of the neural network model ensures that the applied current does not exceed the safety limits.

It should be noted that a similar approximation capability for the output-based deep MPC can be achieved for initial conditions, references, and parameters that fall in the intervals of variation considered during the training phase. These intervals can be set wide enough to account for the entire range of realistic scenarios. To support this claim, the paper provides a statistical analysis comparing the output-based deep MPC profiles with a benchmark over 100 simulations with randomly chosen starting states, references, and parameters. In particular, it emerges that the approximation error for the applied current is, on average, −3.9 mA, with a standard deviation of 487.9 mA, as depicted in a histogram that resembles a normal distribution in Figure 7.

Table 2 summarizes the approximation errors and standard deviations for the state of charge, voltage, and core temperature profiles. The average approximation error for the state of charge profile is $-0.2\times 10^{-3}$, and the standard deviation is $10.1\times 10^{-3}$. The voltage profile has an average error of −0.2 mV, with a standard deviation of 14.4 mV, while the core temperature has an average error of −24.4 mK, with a standard deviation of 401.7 mK. The algorithm has a higher probability of making errors in the approximation of the MPC control law when the system state is near the reference or constraints. Although these errors can cause small oscillations around the reference, they do not affect the stability of the control law. However, including additional samples corresponding to critical points in the dataset could improve the accuracy of the algorithm.

Finally, we note that, although the output-based dMPC does not imply an initial guess for the battery’s states or the battery parameters, a guess for the first $n_{w}$ control action is required, due to the fact that the proposed approach can predict the optimal current only when a window of $n_{w}$ historical measurements is available. Such an applied current guess is computed offline as the average current applied at the beginning of the charging phase according to the available data in the training set.

### 5.2. Implementation Details

The simulations presented in this paper were conducted on a personal computer running Windows 10 with an i7-8750H processor and 16 GB of RAM, and the implementation was carried out using Python 3.7. The deep learning model was created and trained using TensorFlow 2.0, while the equations of the model were integrated via CasADi, a symbolic framework that allows for automatic differentiation. CasADi was also employed to solve the optimal control problem described in Equation (Equation 7).

## 6. Discussion

As stated, the methodology described in this paper is a novel contribution. However, corresponding issues need to be taken into account. Specifically, the aspects mentioned in the following paragraphs may be considered in future research.

The proposed methodology is designed to be robust with respect to uncertainty in the battery parameters. However, it should be highlighted that a high performance can be achieved in a practical scenario only under the assumption that all of the relevant physical phenomena have been considered in the battery model used to generate the synthetic dataset. Although this assumption is more realistic than assuming perfect knowledge of states and parameters, it is important to investigate the consequences of its violation and develop countermeasures to mitigate the effects of potential inaccuracies in the modeling phase.The output-based deep MPC predicts the optimal input for charging a battery based on previous measurements. However, at the start of the charging process, there are only a limited number of previous measurements available, making it challenging to effectively use the proposed algorithm. To address this challenge, a rough guess for the initial control actions is utilized. This guess is computed offline as the average current applied at the beginning of the charging phase, based on the available data in our training set. Unfortunately, if the guess input deviates from the optimal one, the controller’s performance during the first stage of charging may be affected, potentially leading to safety issues. To mitigate this issue, multiple output-based deep MPC algorithms can be employed during the first stage of charging. Each algorithm utilizes a different number of previous measurements, starting from no previous measurements, then gradually incorporating one, two, and so on, until a certain number of measurements are available.

Addressing the aforementioned issues in the future may pave the way to further improve the proposed approach.

## 7. Conclusions

In this paper, the optimal charging of a battery is addressed by exploiting a deep learning-based methodology. This methodology has been designed to be effective even in a realistic scenario in which the battery parameters are unknown and only noisy measurements are available. To the best of the authors’ knowledge, for the first time, the proposed technique relies on recurrent neural networks to approximate the optimal control law of a predictive controller, by using only voltage and temperature data. Firstly, battery charging was formalized as an optimal control problem. Then, a detailed description of the generation of a synthetic dataset for the training phase was provided, highlighting the importance of exploring different battery conditions and parameters. In addition, an analysis was conducted to select the most suitable machine learning model, in order to find a reasonable trade-off between accuracy and computational complexity. Finally, the validation of the proposed algorithm was performed against an ideal predictive controller (with full-state measurability and parameter knowledge) used as a benchmark. The results show that the output-based deep MPC is able to achieve a performance similar to the benchmark without relying on the unrealistic assumption of knowing the battery states and parameters, but only on noisy measurements. In this regard, a statistical analysis was provided to highlight the effectiveness of the proposed approach over 100 different simulations.

## Figures and Tables

**Figure 1 sensors-23-04404-f001:**
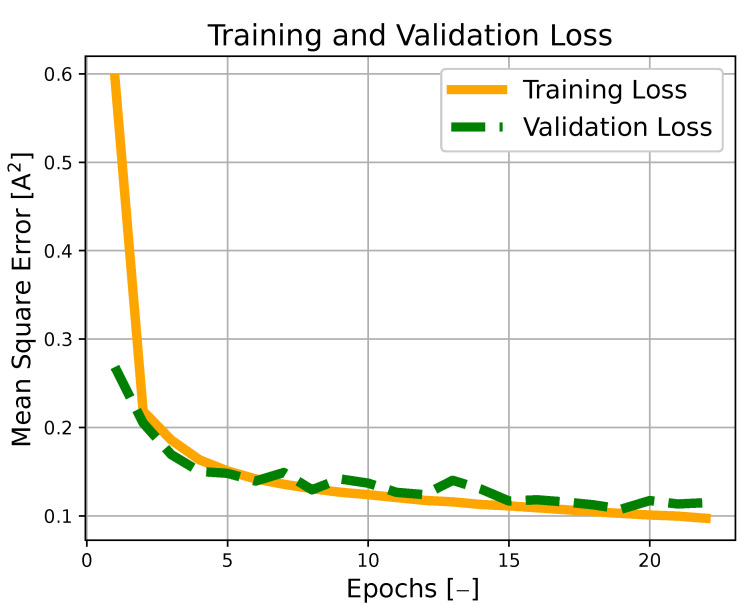
Training and validation loss profiles against epochs for the RNN model with 4 long short-term memory (LSTM) hidden layers (with 128, 64, 32, and 16 neurons each) and 4 fully connected hidden layers (2 of them with 100 neurons each, one with 50 neurons, and the last one with 10 neurons), with a window of historical measurements $n_{w}=20$.

**Figure 2 sensors-23-04404-f002:**
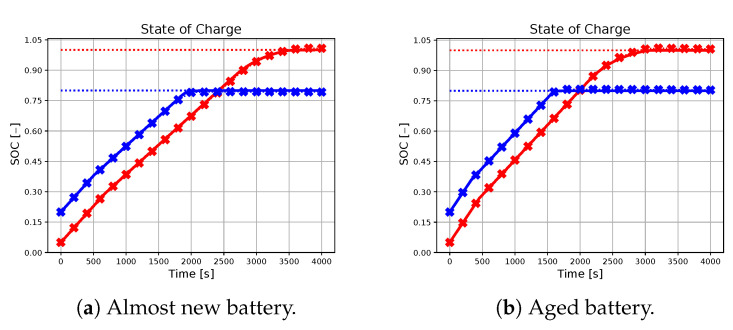
State-of-charge profiles of the two considered methodologies (solid line for the ideal predictive controller and crossed line for the output-based deep MPC), with *sim 1* in blue and *sim 2* in red. The blue and red dotted lines represent the SOC references for the first and second simulations, respectively. The case of a new battery is considered on the left, while an aged battery is considered on the right.

**Figure 3 sensors-23-04404-f003:**
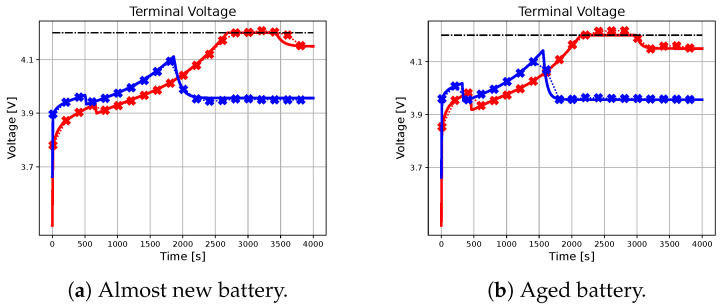
The figure illustrates the profiles of the terminal voltage achieved by the benchmark and the proposed methodology for different initial conditions and battery parameters. The dash-dotted black line represents the upper-bound voltage.

**Figure 4 sensors-23-04404-f004:**
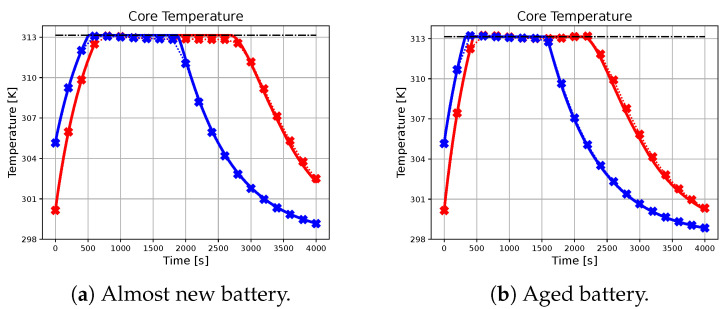
The figure illustrates the profiles of the core temperature achieved by the benchmark and the proposed methodology for different initial conditions and battery parameters. The dash-dotted black line represents the upper-bound for the core temperature.

**Figure 5 sensors-23-04404-f005:**
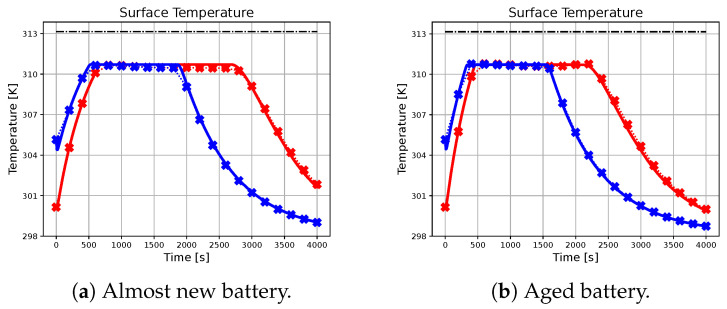
The figure illustrates the profiles of the surface temperature achieved by the benchmark and the proposed methodology for different initial conditions and battery parameters. The dash-dotted black line represents the upper-bound for the surface temperature.

**Figure 6 sensors-23-04404-f006:**
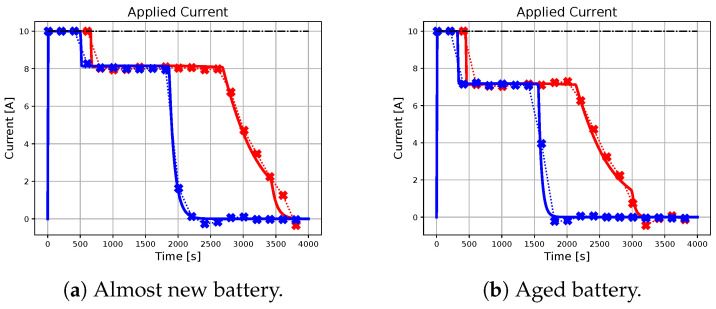
The figure illustrates the profiles of the current applied by the benchmark and the proposed methodology for different initial conditions and battery parameters. The dash-dotted black line represents the upper-bound for the applied current.

**Figure 7 sensors-23-04404-f007:**
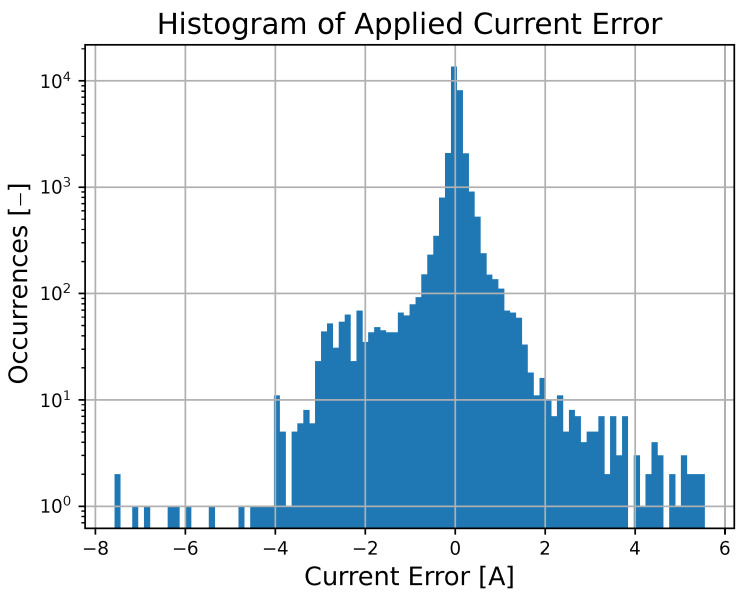
Histogram of the errors observed in over 100 simulations by the output-based deep MPC approach in approximating the ideal predictive control law.

**Table 1 sensors-23-04404-t001:** Mean square error (MSE) over the validation set of the machine learning models considered as possible candidates for the MPC law approximation under the assumption of non-measurable states and unknown model parameters.

Model	Measurements Window ($n_{w}$)	Mean Squared Error
DNN (Dense: 3×100,3×50,3×10)	10	0.256
DNN (Dense: 3×100,3×50,3×10)	15	0.212
DNN (Dense: 3×100,3×50,3×10)	20	0.199
RNN (LSTM: 64,32,16, Dense: 3×100,3×50,3×10)	10	0.202
RNN (LSTM: 64,32,16, Dense: 3×100,3×50,3×10)	15	0.182
RNN (LSTM: 64,32,16, Dense: 3×100,3×50,3×10)	20	0.124
RNN (LSTM: 128,64,32,16, Dense: 2×100,50,10)	10	0.193
RNN (LSTM: 128,64,32,16, Dense: 2×100,50,10)	15	0.133
**RNN (LSTM:** 128,64,32,16 **, Dense:** 2×100,50,10)	**20**	**0.109**

**Table 2 sensors-23-04404-t002:** Statistical description of over 100 simulations of the difference between the trajectory of the model variables obtained with the 2 considered control methodologies.

Statistics	$\mathrm{soc}\;\left[10^{-3}\right]$	V[mV]	$T_{c}\;\left[\mathrm{mK}\right]$
Mean	$\begin{array}{c} -0.2 \end{array}$	$\begin{array}{c} -0.1 \end{array}$	$\begin{array}{c} -24.4 \end{array}$
Standard deviation	$\begin{array}{c} 10.1 \end{array}$	$\begin{array}{c} 14.4 \end{array}$	$\begin{array}{c} 401.7 \end{array}$

## Data Availability

Data sharing is not applicable.

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
