# Peer review of "Optimizing Battery Charging Using Neural Networks in the Presence of Unknown States and Parameters"

_sensors, 2023, doi:10.3390/s23094404_

Round 1
Reviewer 1 Report
The manuscript “Optimizing Battery Charging Using Neural Networks in the Presence of Unknown States and Parameters " is devoted to the calculations for battery charging using neural networks for scenarios where the battery’s internal states cannot be measured and an estimate of the battery parameters is unavailable. The work is devoted to a very relevant topic. A larger statistical analysis of the quality of the model could be done.
I think, this manuscript can be published in the Sensors after minor revision taking into account general recommendation described below:
Self-citation of the Andrea Pozzi author is 23% (articles number 8, 18, 19, 20, 25, 26, 27, 35). It is necessary to reduce this figure to 10-15%.
The topic of the article is relevant. However, this is only the beginning of development - relatively little research has been done. This work cannot make a big contribution to the research area.
According to the authors, this work is the first in the field of neural network-based approach for scenarios where the battery’s internal states cannot be measured and an estimate of the battery parameters is unavailable. The study of the fundamental possibility of solving this problem is the main task of the article.
As additional control methods, we can recommend to increase the literature review of the studies already done. It is unlikely that there are no similar or related works for such a topical topic. It is also necessary to conduct more statistical studies to verify the proposed model.
The conclusions are consistent with the evidence presented. But at the end of the work there is a large plan of additional research that needs to be done. Probably, this article should be considered not as a long article, but as a short message.
It is necessary to reduce the share of authors' self-citations.
Reviewer 2 Report
1. In the introduction, the author focuses on the current research status of MPC. However, electrochemical models as the basis of charging strategies should also be discussed.
2. The presentation in Table 2 is flawed and needs further improvement by the author.
3. Whether the current I(t)in Eq. (1) and Eq. (2) is the same physical quantity? In addition, V is a nonlinear function of time, current and temperature, not just a function of time in Eq. (2).
4. It is necessary to cite relevant literature for the source of Eq. (3).
5. The accuracy of the single particle model in the part.2 is not mentioned, which raises questions about the results of subsequent studies.
6. The conclusion part is too cumbersome, please cut it appropriately.
